# Intrahepatic Duct Incision and Closure for the Treatment of Multiple Cholelithiasis in a Dog

**DOI:** 10.3390/vetsci11080378

**Published:** 2024-08-16

**Authors:** Ji-Hoon Kang, Mi-Young Noh, Hun-Young Yoon

**Affiliations:** 1Deukso Animal Hospital, 92 Deukso-ro, Wabu-eup, Namyangju-si 12210, Gyeonggi-do, Republic of Korea; krkdwlgns@naver.com (J.-H.K.); greenmoon96@hanmail.net (M.-Y.N.); 2Department of Veterinary Surgery, College of Veterinary Medicine, Konkuk University, 120 Neungdong-ro, Gwangjin-gu, Seoul 05029, Republic of Korea; 3KU Center for Animal Blood Medical Science, Konkuk University, 120 Neungdong-ro, Gwangjin-gu, Seoul 05029, Republic of Korea

**Keywords:** multiple cholelithiasis, intrahepatic duct incision, intrahepatic cholelithiasis, canine cholelithiasis

## Abstract

**Simple Summary:**

In veterinary medicine, intrahepatic cholelithiasis is a difficult disease to manage. Previously, treatment methods such as medical management or a liver lobectomy were suggested, although they cannot be used to treat some patients. Here, we describe the use of intrahepatic duct incision and closure to remove most of the cholelithiasis in the hepatic ducts of the left medial and lateral liver lobes in a dog with untreated hypothyroidism and multiple cholelithiasis. To our knowledge, this is the first report of this procedure in veterinary medicine. The findings from this case suggest that this technique can be used as a safe and feasible option in veterinary medicine as well as human medicine. However, because it often cannot remove all intrahepatic cholelithiasis, physicians must be cautious when choosing this treatment method. Although it is not the best way to treat intrahepatic duct cholelithiasis, it can be a new treatment option for patients who cannot receive conventional treatment methods.

**Abstract:**

This report describes the successful intrahepatic duct incision and closure for the treatment of multiple cholelithiasis in a dog with untreated hypothyroidism. A 12-year-old spayed female Spitz dog weighing 11.3 kg was diagnosed with multiple cholelithiasis, and a quadrate liver lobectomy and cholecystectomy were performed. Large gallstones were located in the left liver lobe’s intrahepatic duct distal to the anastomosis of the intrahepatic ducts of the left medial and lateral lobes. The dilated intrahepatic duct was packed off with wet gauze, and incision and closure were performed on the most dilated section, which was proximal to the largest gallstone. After surgery, the patient showed normal liver function and was discharged with normal total bilirubin and C-reactive protein levels. On postoperative day 83, no stones were observed in the dilated common bile duct (CBD), and the degree of dilatation of the CBD had decreased from 9 mm to 4 mm, with no obstructions. Right intrahepatic gallstones were confirmed without dilatation. Hypothyroidism was managed medically. Hepatic duct incision and closure can be performed in dogs with multiple cholelithiasis. Although not the first option, intrahepatic bile duct incision proves to be a new alternative for the successful treatment of cholelithiasis in dogs.

## 1. Introduction

Cholelithiasis in dogs is uncommon and has an incidence of <1% of all biliary system diseases [1], with clinical signs including a loss of appetite, vomiting, pyrexia, abdominal pain, and jaundice [2]. If obstruction occurs in the biliary tract, the disease can progress acutely, and in severe cases, emergency surgery is required. Although cholelithiasis’s exact causes have not yet been identified, biliary tract infections or endocrine diseases, such as hypothyroidism, can cause cholelithiasis [3,4,5]. If intrahepatic cholelithiasis is not treated, acute cholecystitis and cholangitis, acute pancreatitis, bile duct obstruction, gallstone ileus, and gallbladder (GB) rupture can occur [6]. In human medicine, untreated intrahepatic lithiasis can cause cholangitis, cholangiocarcinoma, or sepsis [7].

In cases with cholelithiasis with no other obstructions, the primary treatment options include treatment of the underlying disease and medications such as ursodeoxycholic acid (UDCA), which may help reduce or dissolve cholelithiasis. Surgical treatment is not required if medical management is successful [5,8,9,10]. However, if partial or complete obstructions are present, surgical treatment may be required, with excision of the GB with cholelithiasis or a hepatic lobectomy with intrahepatic cholelithiasis as the options available in veterinary medicine [11,12,13]. If cholelithiasis is present in the common bile duct (CBD), the duodenum is incised, and the stones are retracted into the GB and removed along with the GB. If it is difficult to secure patency through retrograde flushing, patency can be secured through normograde flushing by inserting a soft catheter into the GB [13]. If gallstones are not removed following flushing, choledochotomy can be considered. However, only a few cases regarding treating cholelithiasis involving multiple intrahepatic bile ducts have been reported in veterinary medicine [11,12,14].

Herein, we present a patient with multiple cholelithiasis in the GB and cystic, intrahepatic, and CBDs which was successfully surgically removed. To the best of our knowledge, this is the first report of the surgical treatment of cholelithiasis in the intrahepatic bile duct through intrahepatic bile duct incision in veterinary medicine.

## 2. Case Description

A 12-year-old spayed female Spitz dog weighing 11.3 kg visited the Deukso Animal Hospital for evaluation due to anorexia and lethargy. The patient was previously diagnosed with hypothyroidism and treated medically. However, the patient was lost to follow-up for approximately 10 months, and medical management was discontinued. The patient was mildly depressed and had mild skin icterus. No other remarkable physical examination findings were identified. 

Complete blood count showed mild anemia (hematocrit, 30.2%; reference range [RR], 37.3–61.7%), and serum biochemistry showed severely increased liver enzyme levels. Additionally, increased total bilirubin levels (1.2 mg/dL; RR, 0.0–0.9 mg/dL), decreased total T4 (0.5 μg/dL; RR, 0.8–3.5 μg/dL), hypercholesterolemia, and increased triglyceride levels were observed. Additionally, the C-reactive protein level was also significantly increased (9.8 mg/dL; RR, 0.0–1.0 mg/dL). The electrolyte and coagulation test results were unremarkable.

Abdominal radiography revealed radiopaque material suggestive of cholelithiasis in the liver, GB, and CBD area. Ultrasonography (US) showed a mucocele in the GB (Figure 1a) and hepatolithiasis in the left medial, quadrate, and right liver lobes and the GB (Figure 1b,c). The intrahepatic bile duct was dilated with a diameter of approximately 3.3 mm. Distal CBD dilatation was not observed.

Computed tomography (CT) was performed to determine the size and location of the gallstones and whether there was obstruction or dilatation of the biliary system for surgical planning. The CT scan revealed cholelithiasis in the GB, the left, quadrate, and right liver lobes, and the CBD (Figure 2). Multiple intrahepatic gallstones were observed, and the hepatic duct in the left medial liver lobe was dilated with a diameter of 4 mm. The CBD was also dilated to 7 mm, and two gallstones were also identified. 

After CT, the patient was hospitalized and treated with prophylactic antibiotics and fluid therapy. At this time, anorexia and hypoglycemia were observed and managed with fluid therapy using a lactated Ringer’s solution containing 2.5% dextrose. A hepatic lobectomy of the left medial and quadrate liver lobes and a cholecystectomy with confirmation of the patency of the CBD were planned. Although intrahepatic lithiasis was also identified in the right liver lobe, dilatation of the hepatic bile duct in the right liver lobe was not observed, and the diameters of the intrahepatic gallstones in the right liver lobe were not >3 mm. If the right liver lobe is resected, too much of the hepatic lobe may be resected, and there is a risk that normal liver function will not be maintained after surgery. In addition, after the patient returns to normal function through postoperative treatment of the underlying disease and medical management, a second operation can be planned if necessary. Therefore, a lobectomy of the right lobe of the liver was not performed. Surgery was performed two days after the CT scan for stabilization.

The patient was premedicated with cefazolin (22 mg/kg), famotidine (1 mg/kg), metronidazole (10 mg/kg), meloxicam (0.2 mg/kg), maropitant (1 mg/kg), and butorphanol (0.2 mg/kg). Anesthesia was induced using propofol (5 mg/kg). General anesthesia was maintained using isoflurane in 100% oxygen. Cefazolin was administered 60 min before surgery and repeated every 90 min. During the surgery, the patient’s systolic blood pressure was observed to be low and temporarily dropped to 50 mmHg. A constant rate infusion of dobutamine and vasopressin was administered to address this decrease in blood pressure.

A dilated GB was identified, and multiple intrahepatic lithiasis-filled liver lobes were observed. To identify the hepatic ducts of the other lobes and enlarge the surgical field, the quadrate liver lobe was resected first, and the hepatic duct was subsequently ligated. The ligature was placed as close to the cystic duct as possible, taking care not to leave any gallstones. After the quadrate liver lobectomy, the hepatic ducts of the left medial and lateral liver lobes dilated to 4–5 mm were identified. The largest gallstones were located in the intrahepatic duct of the left liver lobe distal to the anastomosis of the intrahepatic duct of the left medial and lateral liver lobes. No gallstones were observed in the intrahepatic bile duct of the left lateral liver lobe. If a left medial liver lobectomy had been performed as planned, the left lateral liver lobe should also have been resected. If the left medial and lateral liver lobes had been resected together, liver dysfunction could have occurred since the quadrate liver lobe would have been resected and the right liver cholelithiasis treated medically. The surgical plan was thus changed to maximally remove the cholelithiasis in the left medial liver lobe through an intrahepatic incision without resection of the left hepatic lobe.

The dilated hepatic duct was packed off with wet gauze, and an incision was made on the most dilated section of the intrahepatic duct, which was the proximal aspect of the largest gallstone. The incision was performed 5 mm parallel to the intrahepatic duct’s longitudinal axis (Figure 3a). When the intrahepatic duct was incised, the tied suture knot was raised to create tension from the bottom of the left lateral hepatic duct to the top to prevent gallstones from passing into the hepatic duct of the left lateral liver lobe. Immediately after the incision of the hepatic duct, the largest gallstone was removed using vascular DeBakey thumb tissue forceps, and then, the hepatic duct was flushed and suctioned using a 5 to 6 Fr feeding tube and a 0.35 Fr Tomcat catheter with the guide removed. After removal of the intrahepatic cholelithiasis and bile, the hepatic duct was apposed with simple continuous 5-0 monofilament absorbable (polyglyconate) sutures (Figure 3b). Two gallstones in the CBD were removed by retrograde flushing into the GB using duodenal enterotomy, and a cholecystectomy was performed. After confirming hemostasis and no bile leakage from the surgical site, the abdominal cavity was flushed with warm 0.9% normal saline, and the abdominal wall and skin were closed.

After surgery, the patient’s blood glucose levels were continuously monitored until they stabilized, and fluid therapy was adjusted by adding dextrose into the intravenous fluid to maintain glucose levels. Additionally, to prevent infection, the patient received cefazolin, enrofloxacin, and metronidazole. For postoperative care, the patient was administered meloxicam and maropitant for three days to manage pain and prevent nausea and vomiting. Furthermore, the contents within the GB were collected for microbial culture, and the gallstone was subjected to stone analysis.

Recovery from the anesthesia was uneventful, and the cholelithiases of the left liver lobe were completely removed based on radiography and US. Four days postoperatively, total bilirubin levels normalized and their appetite recovered. Fluid therapy was maintained with a lactated Ringer’s solution containing 2.5% dextrose for six days postoperatively. On postoperative day 6, blood glucose levels stabilized, and the fluid therapy was changed to a lactated Ringer’s solution. The patient was discharged seven days postoperatively. Thirteen days postoperatively, liver enzyme levels normalized, and US showed that the dilatation of the left hepatic duct had resolved, with no signs of intrahepatic gallstones. An enlarged cystic duct was identified near the duct at the conjuncture with the left hepatic duct; however, no obstruction was confirmed. No other dilated ducts, except for the cystic duct, were observed. The bile microbial culture was unremarkable, and the gallstones were 100% calcium carbonate on analysis. On postoperative day 25, the CBD was dilated to approximately 9 mm, and stones were found in the dilated CBD. However, there was no duodenal opening obstruction. On postoperative day 183, all of the stones in the dilated CBD were resolved, and the CBD was only dilated to 4 mm, with no obstructions observed. Hypothyroidism was managed medically with levothyroxine, and right intrahepatic gallstones were confirmed without dilatation.

## 3. Discussion

Some studies in veterinary medicine have suggested that the occurrence of cholelithiasis is related to biliary system infections; however, in other studies, this aspect remains unclear [13,15,16,17]. Lee et al. reported the occurrence of cholelithiasis and GB mucoceles due to hypothyroidism [4]. In their study, hematological changes and changes in bile viscosity caused gallstones and GB mucoceles. Additionally, their study reported that patients with high cholesterol, triglycerides, and insulin serum concentrations had a significantly higher cholelithiasis occurrence than in the control group [4]. The odds ratio for cholelithiasis in patients with hypercholesterolemia was 9.72 times higher than that in healthy subjects. In human medicine, a study also showed that cholelithiasis may increase with hypercholesterolemia. However, this is an unlikely etiology considering the composition of cholelithiases in dogs [3,4,5]. In the present case, the patient had chronic untreated hypothyroidism, which resulted in chronic hypercholesterolemia. This caused high cholesterol concentrations in the GB, which may have caused the GB mucoceles. The GB mucoceles may have increased the bile viscosity by delaying bile excretion, which may have increased the concentration of various components and acted as a nidus for stones [5,18]. Experimental studies have reported that microliths occurred when there was CBD obstruction, ultimately resulting in cholelithiasis [19,20]. In our case, CBD cholelithiasis may have occurred due to an unknown cause, resulting in partial obstruction. In addition, this partial obstruction may have caused bile stasis in the intrahepatic duct, resulting in the formation of multiple intrahepatic lithiases. Additionally, the extracted gallstones were composed of 100% calcium carbonate, which is consistent with a previous study in which most components of gallstones were identified as either calcium carbonate or calcium bilirubinate [21]. Since calcium salts can act as nuclei in patients with hypercholesterolemia [13,21], it can be assumed that cholelithiasis occurred in this patient due to untreated hypothyroidism.

Not all patients with cholelithiasis require surgery if they can be treated with UDCA [5,9,10]. Surgical treatment is also not required if patients are asymptomatic and show no signs of obstruction. However, surgical treatment may be considered if there is a possibility of obstruction or if clinical symptoms are caused by the obstruction. CBD patency should be confirmed, and the GB, with or without gallstones, can be removed. If gallstones cannot be removed from the CBD by retrograde or normograde flushing, choledochotomy can be performed [13]. However, in the case of intrahepatic bile duct cholelithiasis, surgical treatment is limited. In veterinary medicine, intrahepatic bile duct cholelithiasis can be removed along with the hepatic lobe when bile stasis occurs [12]. However, few such cases have been reported, and other methods have not yet been reported. In human medicine, a method of removing gallstones through incision of the intrahepatic bile duct has been reported, and stones in the intrahepatic bile duct can be removed using only this treatment method [22,23]. In our case, it was shown that intrahepatic cholelithiasis could be removed by intrahepatic bile duct incision, and a new treatment method was proposed when obstruction occurred due to intrahepatic lithiasis and liver damage was not expected.

In veterinary medicine, the risk factors or complications associated with bile duct incisions remain unknown because cases of hepatic duct incisions have not been reported. In our opinion, to decide on hepatic duct incision, almost all gallstones should be removed through the incision. If the gallstones are not completely removed, residual gallstones can cause obstruction, leading to cholangitis, cholecystitis, pancreatitis, bile leakage, or bile duct rupture [6]. Second, after hepatic duct incision, the remaining liver lobe must be able to function normally. If the affected liver lobe is damaged by lithiasis obstruction and is not expected to function normally after surgery, a liver lobectomy with intrahepatic lithiasis is recommended. However, if multiple liver lobes are affected, and since a liver lobectomy can cause severe liver dysfunction, less affected liver lobe(s) may be considered for incision of the intrahepatic duct. If intrahepatic cholelithiasis does not completely resolve in the remaining liver lobe, a second surgery is required to excise the liver lobe after its regeneration. During the regeneration of the additional liver lobe, smaller cholelithiasis of the intrahepatic duct can be resolved through a dilated CBD. Third, in our case, the degree of hepatic duct dilatation was moderate to severe. If excessive stone removal and flushing had been performed in mildly dilated or normal hepatic ducts, strictures might have occurred because of iatrogenic lumen injury. In addition, closure of the non-dilated duct after incision could have resulted in the narrowing of the lumen, followed by additional obstruction. Similar to a choledochotomy, the diameter of the dilated intrahepatic duct was greater than 4 to 5 mm, and the end of the incised duct was sutured tension-free to prevent stenosis of the hepatic duct. In addition, the hepatic duct was not grossly damaged. If leakage occurred, additional sutures were placed.

If a patient has an underlying disease that is suspected to cause cholelithiasis, treatment of the underlying disease is of utmost importance. In general, cholelithiasis or GB mucoceles can be treated if an underlying disease is present. While this treatment is in progress, medications such as UDCA, silymarin, and S-adenosyl-l-methionine may be helpful. However, surgical treatment is required if there is an obstruction of the biliary tract and clinical symptoms such as icterus. In our case, the patient showed various clinical symptoms, including icterus, due to the obstruction of the intrahepatic duct and CBD that required surgery. The severely affected liver lobe was excised, and the patency of the obstructed CBD was confirmed. After surgery, the underlying disease was treated, and appropriate medications were administered during liver lobe regeneration. During this period, the patient was followed up, and smaller cholelithiasis was excreted through the dilated CBD. However, as the right hepatic lobe cholelithiasis has not been resolved, we believe a second operation may be needed after liver regeneration.

The present case shows that hepatic bile duct incision and closure are safe and feasible surgical procedures for patients with multiple intrahepatic cholelithiases. Notably, a method to remove all intrahepatic gallstones, such as a hepatic lobectomy, may still be a better option than hepatic duct incision. However, if gallstones are present in multiple liver lobes and not all liver lobes can be removed, hepatic duct incision can be a feasible alternative treatment option. When using this surgical procedure, incompletely damaged liver lobes can be preserved for additional liver regeneration. If adequate liver regeneration is achieved, a second surgery can be planned while maintaining liver function. Fortunately, a second surgery may not be necessary if small stones are excreted through the dilated CBD during liver regeneration and if treatment of the underlying disease does not result in additional cholelithiasis. However, further studies are needed to confirm this hypothesis.

## 4. Conclusions 

Cholelithiasis in the intrahepatic duct is challenging. In some cases, there are reports of a liver lobectomy or treatment with medications, but these are not always successful. This is the first case reported in veterinary medicine in which a dilated intrahepatic duct was incised and sutured, showing that successful results can be obtained as in human medicine. New surgical techniques may increase the number of treatment options for removing cholelithiasis within the intrahepatic duct. However, this surgical technique can be chosen only in limited situations when it is absolutely necessary, and additional research is needed.

## Figures and Tables

**Figure 1 vetsci-11-00378-f001:**
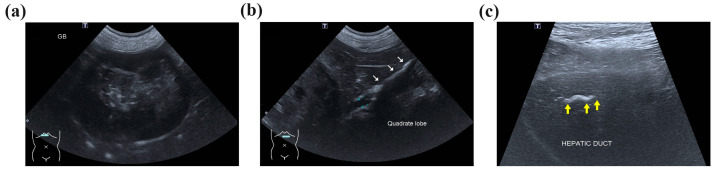
Ultrasonographic imaging showing an enlarged gallbladder (GB) (**a**) due to a GB mucocele and cholelithiasis. The surrounding tissue’s echogenicity was increased, and a small amount of free fluid was also observed. (**b**) The quadrate liver lobe’s hepatic duct was mostly filled with cholelithiases (white arrows) and dilated to 4.2 mm. (**c**) Cholelithiasis was also confirmed in the right liver lobe’s hepatic duct (yellow arrows), but hepatic duct dilatation was not observed. Two cholelithiases were also observed inside the common bile duct, which was dilated to 3.7 mm.

**Figure 2 vetsci-11-00378-f002:**
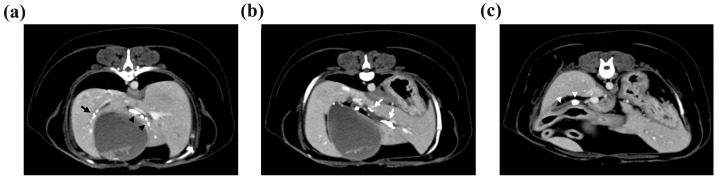
Multiple cholelithiases identified on the computed tomography scan. (**a**) Cholelithiasis in the right liver lobe (black arrow). The diameter of the gallstone was <3 mm. There was no other dilatation of the intrahepatic duct. In the quadrate liver lobe, cholelithiasis was identified from the hilus to the peripheral intrahepatic bile duct (black arrowheads). Mild 3 mm dilatation of the intrahepatic duct was observed. (**b**) Cholelithiasis in the left medial liver lobe. The diameter of the largest gallstone was 7 mm, and the left hepatic duct was dilated to 4 mm (white arrows). (**c**) Two gallstones with 4.8 mm and 9.8 mm diameters were observed in the common bile duct, which was dilated to 7 mm (white arrowheads).

**Figure 3 vetsci-11-00378-f003:**
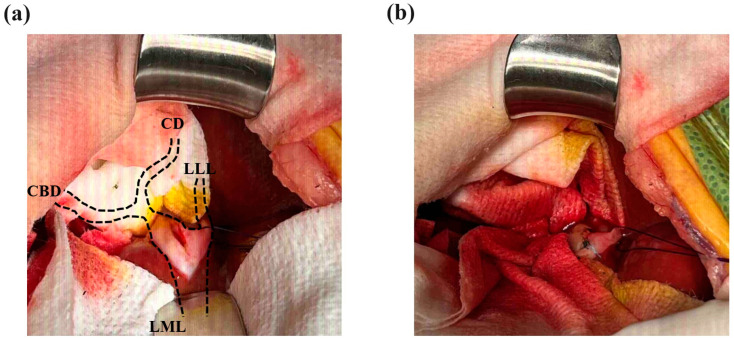
Incision and closure of the intrahepatic bile duct. (**a**) Intrahepatic bile duct obstruction by the largest gallstone occurred after the site of the intrahepatic duct of the left medial liver lobe (LML) and left lateral liver lobe (LLL) was anastomosed. The diameter of the largest gallstone was approximately 7 mm on the CT scan, and it was confirmed to be the same size after surgery. A 5 mm incision was made in the proximal part of the gallstone. (**b**) After flushing the intrahepatic bile duct to remove the gallstones, the incision was closed with a simple continuous 5-0 absorbable suture (white arrow). No gallstones were identified near the common bile duct (CBD) or cystic duct (CD) at the left hepatic duct confluence.

## Data Availability

All study data are presented in the article.

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
