# Peer review of "Intrahepatic Duct Incision and Closure for the Treatment of Multiple Cholelithiasis in a Dog"

_vetsci, 2024, doi:10.3390/vetsci11080378_

Round 1
Reviewer 1 Report
Comments and Suggestions for Authors
Line 190, 191 and 193 "was observed" three times, needs rewording
Line 120... the diamters of the intrahepatic gallstones in the right liver lobe were not < 3 mm. It is no > than 3 mm? no?
Discussion: miniinvasive removal / endoscopically assisted removal?
Author Response
Comments1 : Line 190, 191 and 193 "was observed" three times, needs rewording
Response: Thank you for your careful review of the manuscript. According, the repeated terms have been deleted and the sentence has been revised as follows:
“An enlarged cystic duct was identified near the duct at the conjuncture with the left hepatic duct; however, no obstruction was confirmed.”
Comments2 : Line 120... the diamters of the intrahepatic gallstones in the right liver lobe were not < 3 mm. It is no > than 3 mm? no?
Response: Thank you for your important comment. The diameter of the gallstones did not exceed 3 mm. Therefore, the term “>3 mm” is appropriate. The sentence has been revised as follows:
“…the diameters of the intrahepatic gallstones in the right liver lobe were not >3 mm.”
Comments3 : Discussion: miniinvasive removal / endoscopically assisted removal?
Response: I understand that endoscopic-assisted calculi removal in dogs is not yet recommended.

Reviewer 2 Report
Comments and Suggestions for Authors
The reviewer would like to congratulate the authors on a job well done. Although he has no major comments or modifications to make, I would suggest that the images in the article be presented in a larger size for better viewing.
Author Response
comments1 : The reviewer would like to congratulate the authors on a job well done. Although he has no major comments or modifications to make, I would suggest that the images in the article be presented in a larger size for better viewing.
response1 : Thank you for your kindness. The surgery photos were not taken in high definition at the time, so there are no photos of better quality than the current photos. I apologize for this situation.

Reviewer 3 Report
Comments and Suggestions for Authors
Dear authors,
Thank you for submitting this article for review. This case report is interesting and will contribute to veterinarians’ everyday clinical practice. Before acceptance, I have some considerations.
Title
Just a tip for the title as "incision and closure" sounds a little bit weird: “Intrahepatic duct dissection for removal of multiple cholelithiasis…”; feel free to ignore this comment if you don't like.
Simple summary
Line 16 : try with “ to manage” or “to handle” instead of “to treat”
Line You stated “untreated hypothiroydism” but in the case description (line74-75) you state the opposite, please clarify.
Abstract
Line 39: please modify “was confirmed” with “proves to be a new alternative….”
Case description
Line 72: remove the sentence, you state in the statement below
Line 131-132: “patient’s blood pressure was observed to be low and temporarily dropped to 50 mmHg” Hypotension is generally identify as MAP <60 or SAP <100 mmHg, pleas clarify better your intraoperative findings
Line 152: “the suture was raised from the bottom of the left lateral hepatic duct” what suture, it was a pre-applied suture around the duct or a tension suture? Please clarify better this part of the surgical technique.
Line 178-179: why you decide to applied this prophylactic combination of antibiotics in absence of sign of true infection?
Line 196: please modify with “After 183 days from surgery,”
Line 198: “Hypothyroidism was managed medically” what kind of medication you give, pleas add in the text.
Figure 3: pleas provide intraoperative image with a better quality.
Discussion
Line 202: please modify with “however, in other studies this aspect remain unclear”
Line 227: please clarify the statement “possibility of obstruction”, as large stones? Ore signs of obstruction without symptoms?
Comments on the Quality of English Language
The quality of English is adeguate; only a few sentences could be more concise.
Author Response
Title
Comments1:
Just a tip for the title as "incision and closure" sounds a little bit weird: “Intrahepatic duct dissection for removal of multiple cholelithiasis… ”; feel free to ignore this comment if you don't like.
Response: Thank you for your valuable suggestion. As the bile duct was incised and sutured again, the expression “incision and closure” has been used. We could not find an appropriate word for “intrahepatic duct incision,” as no term has yet been defined. Further, in this case report no problems were encountered even when the intrahepatic duct was incised and closed with sutures. Therefore, the terms “incision” and “closure” were included. We have retained the original title.
Simple summary
Comments2:
Line 16: try with “to manage” or “to handle” instead of “to treat”
Response: Thank you for your insightful suggestion. Accordingly, the term “treat” was revised to “manage” for better clarity.
Comments3:
Line You stated “untreated hypothiroydism” but in the case description (line74-75) you state the opposite, please clarify.
Response: Thank you for your pertinent query. The patient was diagnosed with hypothyroidism 10 months before multiple intrahepatic duct cholelithiasis was identified and treatment was initiated for the same. However, follow-up was lost, and medical management was not performed. After hypothyroidism was diagnosed, treatment was started, but treatment was discontinued; therefore, “untreated hypothyroidism” has been mentioned. Accordingly, “and medical management was discontinued” has been added to the manuscript for clarity .
Abstract
Comments4:
Line 39: please modify “was confirmed” with “proves to be a new alternative…” .”
Response: Thank you for your valuable suggestion. Accordingly, the term “was confirmed” has been revised to "proves to be a new alternative..."
Case description
Comments5:
Line 72: remove the sentence, you state in the statement below
Response: Thank you for your important suggestion. This sentence was included to avoid any ethical issues. As per your suggestion, the sentence has been deleted.
Comments6:
Line 131-132: “patient’s blood pressure was observed to be low and temporarily dropped to 50 mmHg” Hypotension is generally identify as MAP <60 or SAP <100 mmHg, pleas better clarify your intraoperative findings
Response: Thank you for your careful review. The sentence has been revised as follows:
"During the surgery, patient’s systolic blood pressure was observed to be low and temporarily dropped to 50 mmHg."
Comments7:
Line 152: “the suture was raised from the bottom of the left lateral hepatic duct” what suture, it was a pre-applied suture around the duct or a tension suture? Please better clarify this part of the surgical technique.
Response: Thank you for your important question. The purpose of using a suture in this surgery was to prevent damage to the bile duct and to temporarily occlude the bile duct by lifting the suture located nearby without tying it to create tension. We have revised the sentence for clarity as follows:
"When the intrahepatic duct was incised, the suture not tied was raised for tension from the bottom of the left lateral hepatic duct to the top to prevent gallstones from passing into the hepatic duct of the left lateral liver lobe."
Comments8:
Line 178-179: why you decide to apply this prophylactic combination of antibiotics in absence of sign of true infection?
Response: Thank you for your pertinent query. During the process of incising the bile duct, the dilated hepatic duct was packed off with wet gauze to prevent bile leakage. This could have caused infection in the surrounding area . The inflammatory reaction due to infection around the bile duct could possibly have caused obstruction or stenosis of the bile ducts; hence, antibiotics were used until the results of bile microbial culture tests were obtained.
Comment9:
Line 196: please modify with “After 183 days from surgery,”
Response: Thank you for your important comment. Accordingly, this part has been corrected as per your valuable suggestion.
Comments10:
Line 198: “Hypothyroidism was managed medically” what kind of medication you give, pleas add in the text.
Response: Thank you for your important question. The patient was prescribed levothyroxine, and the dosage of the drug was determined according to the patient’s clinical symptoms and hormone concentration test results. This part has been revised to "Hypothyroidism was managed medically with levothyroxine..."
Comments11 :
Figure 3: pleas provide intraoperative image with a better quality.
Response: Thank you for your valuable suggestion. The intraoperative photos were not taken in high definition during the surgery; hence, photos of better quality are not available. We apologize for this situation .
Discussion
Comments12:
Line 202: please modify with “however, in other studies this aspect remains unclear”
Response: Thank you for your valuable suggestion. Accordingly, the necessary correction has been done.
Comments13:
Line 227: please clarify the statement “possibility of obstruction”, as large stones? Ore signs of obstruction without symptoms?
Response: Thank you for your pertinent queries. “Possibility of obstruction” explains all types of obstruction, including obstruction caused by multiple small gallstones or obstruction caused by large gallstones. Even if there are no clinical symptoms, in cases where bile duct dilatation due to obstruction is confirmed, surgical treatment may be considered, and surgery may be performed. For clarity, “required” has been revised to “considered.”

Reviewer 4 Report
Comments and Suggestions for Authors To begin with, I wanted to congratulate the authors for preparing this, we all know that it is not easy to describe and publish a clinical case while carrying out your daily clinical work. The work in general is well structured and clinically well carried out, however, I would like to provide some considerations. The first of them is about the follow-up of the case. No other CT or advanced imaging tests were performed? If it has been done, you should add more information about it. On the other hand, since the authors describe a similar case (published in 2023) (quote number 12), why did they not act in the same way in that previous case? Furthermore, it does not make sense to include bibliographic citations from not so important journals (as number 3, 7 or 12, for instance), and not add this one, for example: - Allan F, Watson PJ, McCallum KE. Clinical features and outcomes in 38 dogs with cholelithiasis receiving conservative or surgical management. J Vet Intern Med. 2021 Nov;35(6):2730-2742. In any case, except for these minimal considerations, the work seems to me to have clinical relevance and provides novelties in the treatment of this pathology, so it should be taken into account for its publication.
Author Response
Comments1 : To begin with, I wanted to congratulate the authors for preparing this, we all know that it is not easy to describe and publish a clinical case while carrying out your daily clinical work. The work in general is well structured and clinically well carried out, however, I would like to provide some considerations. The first of them is about the follow-up of the case. No other CT or advanced imaging tests were performed? If it has been done, you should add more information about it. On the other hand, since the authors describe a similar case (published in 2023) (quote number 12), why did they not act in the same way in that previous case? Furthermore, it does not make sense to include bibliographic citations from not so important journals (as number 3, 7 or 12, for instance), and not add this one, for example: - Allan F, Watson PJ, McCallum KE. Clinical features and outcomes in 38 dogs with cholelithiasis receiving conservative or surgical management. J Vet Intern Med. 2021 Nov;35(6):2730-2742. In any case, except for these minimal considerations, the work seems to me to have clinical relevance and provides novelties in the treatment of this pathology, so it should be taken into account for its publication.
Response: We wish to express our appreciation for the assessment of our manuscript and insightful comments and suggestions. Regarding this case, we did not perform advanced imaging tests during the follow-up because the owner refused to consent. Therefore, only ultrasonography was performed.
Regarding the similar case published in 2023, the case was considered to be inoperable, considering the size of the patient, location of the gallstones, and location of the dilated bile duct; hence, the procedure was not performed at that time.
Regarding references, we have included References 3, 7, and 12 because these articles contained the necessary information for writing this manuscript. The article by Allan F et al. (2021) was excluded from the process of selecting references as it did not contain the necessary information required for this manuscript. However, as per your valuable suggestion we will consider it while writing the next case report.

Round 2
Reviewer 3 Report
Comments and Suggestions for Authors
Dear authors, thank you for your comments and changes to the manuscript, it can now be considered for publication.